# Self-Supervised Learning by Cross-Modal Audio-Video Clustering

**Humam Alwassel**[1][*]
humam.alwassel@kaust.edu.sa

**Dhruv Mahajan**[2]
dhruvm@fb.com

**Bruno Korbar**[2]
bkorbar@fb.com

**Lorenzo Torresani**[2]
torresani@fb.com

**Bernard Ghanem**[1]
bernard.ghanem@kaust.edu.sa

**Du Tran**[2]
trandu@fb.com

[1]King Abdullah University of Science and Technology (KAUST)   [2]Facebook AI
http://humamalwassel.com/publication/xdc

## Abstract

Visual and audio modalities are highly correlated, yet they contain different information. Their strong correlation makes it possible to predict the semantics of one from the other with good accuracy. Their intrinsic differences make cross-modal prediction a potentially more rewarding pretext task for self-supervised learning of video and audio representations compared to within-modality learning. Based on this intuition, we propose *Cross-Modal Deep Clustering (XDC)*, a novel self-supervised method that leverages unsupervised clustering in one modality (*e.g.*, audio) as a supervisory signal for the other modality (*e.g.*, video). This cross-modal supervision helps XDC utilize the semantic correlation and the differences between the two modalities. Our experiments show that XDC outperforms single-modality clustering and other multi-modal variants. XDC achieves state-of-the-art accuracy among self-supervised methods on multiple video and audio benchmarks. Most importantly, our video model pretrained on large-scale unlabeled data significantly outperforms the same model pretrained with full-supervision on ImageNet and Kinetics for action recognition on HMDB51 and UCF101. To the best of our knowledge, XDC is the first self-supervised learning method that outperforms large-scale fully-supervised pretraining for action recognition on the same architecture.

## 1   Introduction

Do we need to explicitly name the actions of "laughing" or "sneezing" in order to recognize them? Or can we learn to visually classify them without labels by associating characteristic sounds with these actions? Indeed, a wide literature in perceptual studies provides evidence that we rely heavily on hearing sounds to make sense of actions and dynamic events in the visual world. For example, objects moving together are perceived as bouncing off each other when the visual stimulus is accompanied by a brief sound [55], and the location and timing of sounds are leveraged as important cues to direct our spatiotemporal visual attention [19, 42]. The influence of hearing sounds in visual perception is also suggested by perceptual studies showing that individuals affected by profound deafness exhibit poorer visual perceptual performance compared to age-matched hearing controls [11, 39].

In this work, we investigate the hypothesis that spatiotemporal models for action recognition can be reliably pretrained from *unlabeled* videos by capturing cross-modal information from audio and video. The motivation for our study stems from two fundamental challenges facing a fully-supervised

---

[*]Work done during an internship at Facebook AI

line of attack to learning video models. The first challenge is the exorbitant cost of scaling up the size of manually-labeled video datasets. The recent creation of large-scale action recognition datasets [5, 15, 25, 26] has undoubtedly enabled a major leap forward in video models accuracies. However, it may be argued that additional significant gains by dataset growth would require scaling up existing labeled datasets by several orders of magnitude. The second challenge is posed by the unclear definition of suitable label spaces for action recognition. Recent video datasets differ substantially in their label spaces, which range from sports actions [25] to verb-noun pairs for kitchen activities [7]. This suggests that the definition of the "right" label space for action recognition, and more generally for video understanding, is still very much up for debate. It also implies that finetuning models pretrained on large-scale labeled datasets is a suboptimal proxy for learning models for small- or medium-size datasets due to the label-space gap often encountered between source and target datasets.

In this paper, we present three approaches for training video models from self-supervised audio-visual information. At a high-level, the idea behind all three frameworks is to leverage one modality (say, audio) as a supervisory signal for the other (say, video). We posit that this is a promising avenue because of the simultaneous synergy and complementarity of audio and video: correlations between these two modalities make it possible to perform prediction from one to the other, while their intrinsic differences make cross-modal prediction an enriching self-supervised task compared to within-modality learning. Specifically, we adapt the single-modality DeepCluster work of Caron *et al.* [6] to our multi-modal setting. DeepCluster was introduced as a self-supervised procedure for learning image representation. It alternates between unsupervised clustering of image features and using these cluster assignments as pseudo-labels to revise the image representation. In our work, the clusters learned from one modality are used as pseudo-labels to refine the representation for the other modality. In two of our approaches—Multi-Head Deep Clustering (MDC) and Concatenation Deep Clustering (CDC)—the pseudo-labels from the second modality are *supplementary*, *i.e.*, they complement the pseudo-labels generated in the first modality. The third approach—Cross-Modal Deep Clustering (XDC)—instead uses the pseudo-labels from the other modality as an *exclusive* supervisory signal. This means that in XDC, the audio clusters drive the learning of the video representation and vice versa. Our experiments support several interesting conclusions:

- All three of our cross-modal methods yield representations that generalize better to the downstream tasks of action recognition and audio classification, compared to their within-modality counterparts.
- XDC (*i.e.*, the cross-modal deep clustering relying on the other modality as an exclusive supervisory signal) outperforms all the other approaches. This underscores the complementarity of audio and video and the benefits of learning label-spaces across modalities.
- Self-supervised cross-modal learning with XDC on a large-scale video dataset yields an action recognition model that achieves higher accuracy when finetuned on HMDB51 or UCF101, compared to that produced by fully-supervised pretraining on Kinetics. To the best of our knowledge, this is the first method to demonstrate that self-supervised video representation learning outperforms large-scale fully-supervised pretraining for action recognition. Moreover, unlike previous self-supervised methods that are only pretrained on curated data (*e.g.*, Kinetics [26] without action labels), we also report results of XDC pretrained on a large-scale uncurated video dataset.

## 2   Related work

**Early unsupervised representation learning.** Pioneering works include deep belief networks [20], autoencoders [21, 64], shift-invariant decoders [51], sparse coding algorithms [32], and stacked ISAs [31]. While these approaches learn by reconstructing the input, our approach learns from a self-supervised pretext task by generating pseudo-labels for supervised learning from unlabeled data.

**Self-supervised representation learning from images and videos.** Several pretext tasks exploit image spatial context, *e.g.*, by predicting the relative position of patches [8] or solving jigsaw puzzles [40]. Others include creating image classification pseudo-labels (*e.g.*, through artificial rotations [13] or clustering features [6]), colorization [77], inpainting [46], motion segmentation [45], and instance counting [41]. Some works have extended image pretext tasks to video [27, 68, 75]. Other video pretext tasks include frame ordering [9, 33, 38, 74], predicting flow or colors [30, 67], exploiting region correspondences across frames [22, 23, 71, 72], future frame prediction [35, 36, 57, 65, 66], and tracking [73]. Unlike this prior work, our model uses two modalities: video and audio.

**Cross-modal learning and distillation.** Several works [2, 16] train a fully-supervised encoder on one modality and distill its discriminative knowledge to an encoder of a different modality. Other works learn from unlabeled data for a specific target task [78, 53]. Unlike these methods, our work is purely self-supervised and aims at learning representations that transfer well to a wide range

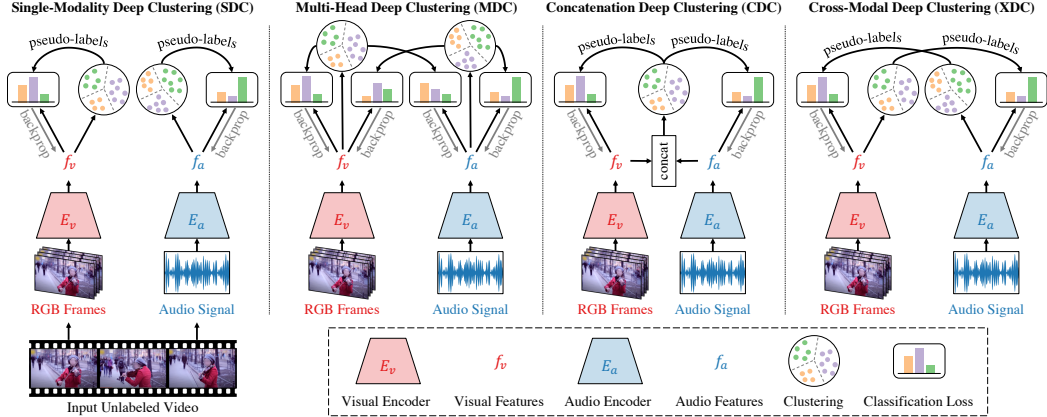

Figure 1: **Overview of our framework.** We present Single-Modality Deep Clustering (SDC) baseline vs. our three multi-modal deep clustering models: Multi-Head Deep Clustering (MDC), Concatenation Deep Clustering (CDC), and Cross-Modal Deep Clustering (XDC). The video and audio encoders ($E_v$ and $E_a$) map unlabeled videos to visual and audio features ($f_v$ and $f_a$). These features, or their concatenations, are clustered using $k$-means. The cluster assignments are then used as pseudo-labels to train the encoders. We start with randomly-initialized encoders, then alternates between clustering to generate pseudo-labels and training to improve the encoders. The four models employ different ways to cluster features and generate self-supervision signals. Illustration video is from [60].

of downstream tasks. Previous cross-modal self-supervised methods most relevant to our work include audio-visual correspondence [1], deep aligned representations [3], audio-visual temporal synchronization [28, 43], contrastive multiview coding [62], and learning image representations using ambient sound [44]. While [1, 3, 44, 62] use only a single frame, we use a video clip. Unlike our method, [44] clusters handcrafted audio features and does not iterate on the pseudo-labels. [28, 43] require constructing positive/negative examples for in-sync and out-of-sync video-audio pairs. This sampling strategy makes these approaches more difficult to scale compared to ours, as many potential out-of-sync pairs can be generated, yielding largely different results depending on the sampling choice [28]. Recent works, such as MIL-NCE [37] and CBT [61], learn from unlabeled instructional videos using text from ASR, while our approach makes use of the audio signal instead.

## 3 Technical approach

Here, we briefly discuss previous work on single-modality deep clustering in images [6]. Then, we introduce our three multi-modal deep clustering frameworks for representation learning (Figure 1).

### 3.1 Single-modality deep clustering

Caron *et al.* [6] proposed DeepCluster for self-supervised representation learning from images. DeepCluster iteratively clusters deep features from a single-modality encoder, and then uses the cluster assignments to train the same encoder to improve its representation. Inspired by the simplicity of this work, our paper studies deep clustering in the large-scale multi-modal setting. For completeness, we summarize DeepCluster details. Let $X$ be the set of unlabeled inputs (*e.g.*, images), $E$ be an encoder that maps an input $x \in X$ to a deep feature vector $f \in \mathbb{R}^d$. DeepCluster iterates between clustering the features $F = \{f = E(x) \mid x \in X\}$ and discriminative training to improve $E$ using the clustering assignments as pseudo-labels. The process starts with a randomly-initialized $E$, and only the weights of the classification `fc`-layer are reset between clustering iterations when the supervision-taxonomy is switched. DeepCluster uses a 2D CNN (*e.g.* ResNet-50) for $E$ and clusters the features after each epoch using $k$-means. We refer to DeepCluster as **Single-Modality Deep Clustering (SDC)**.

### 3.2 Multi-modal deep clustering

Contrary to the single-modality case, there exist multiple encoders in a multi-modal setting, each of which encodes a different modality of the input. In our paper, we consider two modalities, the visual and the audio modalities from the unlabeled training video clips. In particular, let $X$ be the set of unlabeled video clips, and $E_v$ and $E_a$ be the visual and audio encoders, respectively. Let $F_v = \{f_v = E_v(x) \in \mathbb{R}^{d_v} \mid x \in X\}$ and $F_a = \{f_a = E_a(x) \in \mathbb{R}^{d_a} \mid x \in X\}$ be the set of visual

and audio deep features produced by the two encoders, respectively. There are different ways we can adapt the deep clustering framework to a multi-modal input. We describe three approaches (MDC, CDC, and XDC) by detailing the steps taken at each deep clustering iteration. Refer to the *supplementary material* for the implementation differences between SDC and our three approaches.

**Multi-Head Deep Clustering (MDC).** This model builds on SDC by adding a second classification head supervised by the other modality. Thus, in this model, each encoder has two classification heads. At each deep clustering iteration, MDC uses the cluster assignments of $F_v$ as pseudo-labels for one head and that of $F_a$ as pseudo-labels for the other head. Thus, each encoder needs to predict the cluster assignments of its own modality (as in SDC), but also those generated by the other modality.

**Concatenation Deep Clustering (CDC).** This model performs clustering of joint visual and audio features. Specifically, at each deep clustering iteration, CDC clusters vectors obtained by concatenating the visual and audio feature vectors, separately $l_2$-normalized. Then, it uses the resulting cluster assignments as pseudo-labels to update the weights of both $E_v$ and $E_a$.

**Cross-Modal Deep Clustering (XDC).** Each encoder in this model relies exclusively on the clusters learned from the other modality as the supervisory signal. At each deep clustering iteration, XDC clusters the audio deep features, $F_a$, and uses their cluster assignments as pseudo-labels to train the visual encoder, $E_v$. Vice versa, XDC supervises $E_a$ with the cluster assignments of $F_v$.

## 4 Experiments

### 4.1 Experimental setup

**Pretraining datasets.** We use four datasets: Kinetics [26], AudioSet [10], IG-Kinetics [12], and IG-Random, which have 240K, 2M, 65M, and 65M training videos, respectively. As our approach is self-supervised, thus the labels from the first three datasets are **not used** during pretraining. While Kinetics and AudioSet are supervised benchmarks for action recognition and audio classification, IG-Kinetics is a weakly-supervised dataset collected from a social media website using tags related to Kinetics actions. IG-Random is an *uncurated* dataset of random videos from the same website. Videos are 10-second long in Kinetics and AudioSet and 10-to-60-second long in IG-Kinetics and IG-Random. We filter out around 7K Kinetics videos that have no audio. Furthermore, we randomly sample 240K videos from AudioSet and denote this subset as AudioSet-240K. We generate this subset to have AudioSet data of the same size as Kinetics, in order to study the effects of pretraining with the same data size but on a different data distribution and domain.

**Downstream datasets.** We evaluate our pretraining performance on three downstream benchmarks: UCF101 [56], HMBD51 [29], and ESC50 [48], which have 13K, 7K, and 2K examples from 101, 51, and 50 classes, respectively. UCF101 and HMDB51 are action recognition datasets, while ESC50 is a sound classification dataset. UCF101 and HMDB51 have 3 official train/test splits, while ESC50 has 5 splits. We conduct our ablation study (Subsection 4.2) using split-1 of each dataset. We also report our average performance over all splits when we compare with state-of-the-art methods in Section 6.

**Baselines.** We consider two baselines: *Scratch* and *Supervised Pretraining (Superv)*. The first is a randomly-initialized model trained from scratch directly on the downstream task, while the second is a model pretrained in a supervised fashion on a large labeled dataset (*e.g.*, Kinetics) and then finetuned on the downstream task. We note that these two baselines are commonly regarded as the lower and upper bounds to gauge the quality of self-supervised representation learning methods [1, 28].

**Backbone architectures.** We employ R(2+1)D [63] and ResNet [18] as $E_v$ and $E_a$, respectively. $E_v$'s input is a $3{\times}L{\times}H{\times}W$ clip, where 3 refers to the RGB channels, $L$ is the number of frames, and $H$ and $W$ are the frame height and width. $E_a$'s input is a $Q{\times}P$ spectrogram image extracted from the audio signal, where $Q$ is the number of MEL filters and $P$ is the number of audio frames.

**Pretraining and evaluation details.** We choose the 18-layer variants of R(2+1)D and ResNet encoders. We use clips of $L{=}8$ frames for pretraining and finetuning our visual encoder $E_v$. We scale frames such that the smallest dimension is 256 pixels and then random crop images of size $224{\times}224$. We extract video clips at 30 fps and employ temporal jittering during training. For the audio input, we sample 2 seconds and use $Q{=}40$ MEL filters and $P{=}100$ audio frames. For inference on the downstream tasks, we uniformly sample 10 clips per testing example and average their predictions to make a video-level prediction. We use only one crop per clip: the center $8{\times}224{\times}224$ crop for video and the full $40{\times}100$ crop for audio. We provide more details in the *supplementary material*.

Table 1: **Single-modality vs. multi-modal deep clustering.** We compare the four self-supervised deep clustering models (Section 3) and the three baselines: Scratch, Supervised Pretraining (Superv), and same-modality-XDC (XDC with two encoders defined on the same modality). Models are pretrained via self-supervision on Kinetics and fully finetuned on each downstream dataset. We report the top-1 accuracy on split-1 of each dataset. All multi-modal models significantly outperform the single-modality deep clustering model. We mark in bold the best and underline the second-best models.

| Dataset | Scratch | Superv | SDC | MDC | CDC | XDC | same-modality-XDC | |
|---------|---------|--------|-----|-----|-----|-----|-------------------|-------------------|
| | | | | | | | 2 visual encoders | 2 audio encoders |
| UCF101 | 54.5 | 90.9 | 61.8 | 68.4 | 72.9 | **74.2** | 61.3 | N/A |
| HMDB51 | 24.1 | 58.0 | 31.4 | 37.1 | 37.5 | **39.0** | 30.5 | N/A |
| ESC50 | 54.3 | 82.3 | 66.5 | 70.3 | 74.8 | **78.0** | N/A | 66.0 |

## 4.2 Ablation study

**Study 1: Single-modality vs. multi-modal deep clustering.** This experiment compares the four models presented in Section 3. We pretrain SDC, MDC, CDC, and XDC on Kinetics and report their performance on the downstream tasks in Table 1. To better understand XDC, we also conduct a new set of baselines, called same-modality-XDC, where XDC is trained with two encoders defined on the *same* modality (either visual or audio). Note that all models in this ablation study use the same visual and audio encoders and only differ in the way they use self-supervision. It takes on average 5 to 6 deep clustering iterations for these models to converge. *Observations:* **(I)** The four self-supervised deep clustering models outperform the Scratch baseline on all downstream benchmarks. This shows that our self-supervised pretraining is effective and generalizes well to multiple tasks. **(II)** All multi-modal models (MDC, CDC, and XDC) significantly outperform SDC by up to 12.4%, 7.6%, and 11.5% on UCF101, HMDB51, and ESC50, respectively. This validates the importance of multi-modal modeling compared to single-modality. **(III)** XDC achieves the best performance across all tasks. What distinguishes XDC from the other models is that each modality encoder in XDC is self-supervised purely by the signal from the other modality. The encoders in CDC, MDC, and SDC all employ a self-supervision signal coming from the same modality. Thus, this suggests that encoders learn better when purely supervised by a different modality. We provide the following intuition on why XDC is better than CDC and MDC. XDC groups samples together when they are similar in one of the two modalities (video to supervise the audio encoder, audio to supervise the visual encoder). Instead, CDC groups samples together only if they are similar according to both the audio *and* the video modality (to supervise both encoders). Thus, XDC visual and audio clusters allow for more diversity than those of CDC. We hypothesize that this diversity allows XDC to learn richer representations, which translates into better performance on the downstream tasks. Also, recent work [70] has shown that models trained on different modalities learn and generalize at different speeds, and that training them jointly (as done in MDC which uses two-modality heads) is sub-optimal. We believe that this could contribute to MDC performing worse than XDC, which optimizes for each modality independently. **(IV)** The same-modality-XDC baselines perform similarly to SDC and are 8-12% worse than multi-modal-XDC. This suggests that cross-modality provides a superior supervisory signal for self-supervised learning and that multi-modal-XDC is the best model not because of its optimization strategy but rather because of the use of the other modality for pseudo-labeling. Given the results of this study, we opt to use only XDC in the rest of the experiments. Finally, to show that XDC works for different backbones, we re-do Study 1 with ResNet3D in the *supplementary material*.

**Study 2: The number of $k$-means clusters.** This study explores the effects of changing the hyperparameter $k$ in $k$-means clustering. We pretrain XDC on three datasets, Kinetics, AudioSet-240K, and AudioSet, using $k$=64, 128, 256, 512, and 1024 clusters (Table 2). *Observations:* **(I)** The best $k$ value is not sensitive to the number of semantic labels in the downstream datasets. For example, HMDB51 and ESC50 have about the same number of labels but different best $k$ value. **(II)** Similarly, the best $k$ value seems uncorrelated with the number of original semantic labels of the pretraining dataset, *e.g.* 400 in Kinetics. We reiterate here that our approach is self-supervised and **does not use** the labels of the pretraining dataset. **(III)** The best $k$ value tends to get larger as the pretraining data size increases. For example, the best $k$ for HMDB51 shifts from 128 to 256 when moving from pretraining on AudioSet-240K to the full AudioSet. We hypothesize that there is a more diverse sample set to cluster when the pretraining data size increases. Thus, we can have more fine-grained clusters (higher $k$) and make our self-supervised classification problem harder. This aligns with previous self-supervised works [14, 28] that showed benefits from making the pretext task harder.

Table 2: **The number of clusters** ($k$). We show the effect of the number of $k$-means clusters on XDC performance. XDC is pretrained on three large datasets, and then fully finetuned on three downstream tasks. We report the top-1 accuracy on split-1. The best $k$ value increases as the size of the pretraining dataset increases.

| Pretraining Dataset | Downstream Dataset | $k$ | | | | |
|---|---|---|---|---|---|---|
| | | 64 | 128 | 256 | 512 | 1024 |
| Kinetics (240K videos) | UCF101 | 73.8 | 73.1 | **74.2** | 74.0 | 72.6 |
| | HMDB51 | 36.5 | **39.0** | 38.3 | 37.7 | 37.7 |
| | ESC50 | **78.0** | 76.3 | 75.0 | 74.5 | 71.5 |
| AudioSet-240K (240K videos) | UCF101 | **77.4** | 77.2 | 76.7 | 77.1 | 75.3 |
| | HMDB51 | 41.3 | **42.6** | 41.6 | 40.6 | 40.7 |
| | ESC50 | **78.5** | 77.8 | 77.3 | 76.8 | 73.5 |
| AudioSet (2M videos) | UCF101 | 84.1 | 84.3 | **84.9** | 84.4 | 84.2 |
| | HMDB51 | 47.4 | 47.6 | **48.8** | 48.5 | 48.4 |
| | ESC50 | 84.8 | **85.8** | 85.0 | 84.5 | 83.0 |

Table 3: **Pretraining data type and size.** We compare XDC pretrained on five datasets vs. fully-supervised pretrained baselines (Superv). XDC significantly outperforms fully-supervised pretraining on HMDB51.

| | Pretraining | | Downstream Dataset | | |
|---|---|---|---|---|---|
| Method | Dataset | Size | UCF101 | HMDB51 | ESC50 |
| Scratch | None | 0 | 54.5 | 24.1 | 54.3 |
| Superv | ImageNet | 1.2M | 79.9 | 44.5 | NA |
| Superv | Kinetics | 240K | 90.9 | 58.0 | 82.3 |
| Superv | AudioSet-240K | 240K | 76.6 | 40.8 | 78.3 |
| Superv | AudioSet | 2M | 84.0 | 53.5 | **90.3** |
| XDC | Kinetics | 240K | 74.2 | 39.0 | 78.0 |
| XDC | AudioSet-240K | 240K | 77.4 | 42.6 | 78.5 |
| XDC | AudioSet | 2M | 84.9 | 48.8 | 85.8 |
| XDC | IG-Random | 65M | 88.8 | 61.2 | 86.3 |
| XDC | IG-Kinetics | 65M | **91.5** | **63.1** | 84.8 |

**Study 3: Pretraining data type and size.** Here, we investigate the effects of two pretraining characteristics: data size and type. To this end, we pretrain XDC on Kinetics (240K examples), AudioSet-240K (240K examples), AudioSet (2M examples), IG-Kinetics (65M examples), and IG-Random (65M examples). Kinetics and IG-Kinetics videos are collected originally for activity recognition, while AudioSet videos are aimed for audio event classification. IG-Random is an uncurated/unsupervised dataset. In addition to video datasets, we also experiment with ImageNet to understand how much action recognition benefits from supervised pretraining on object classification. For ImageNet, we inflate the images into static video clips (repeating the same frame) and pretrain our video model on this dataset. Table 3 presents the results of this study. ***Observations:*** **(I)** XDC improves across all three downstream tasks as the pretraining data size increases. For example, XDC on HMDB51 improves by $9.8\%$, $22.2\%$, and $24.1\%$ when pretrained on AudioSet, IG-Random, and IG-Kinetics, respectively, compared to the results when pretrained on Kinetics. **(II)** XDC outperforms Kinetics fully-supervised pretraining by $5.1\%$ on HMDB51 and by $0.6\%$ on UCF101. To the best of our knowledge, XDC is the first method to demonstrate that self-supervision can outperform large-scale full-supervision in representation learning for action recognition. **(III)** The performance of the fully-supervised pretrained model is influenced by the taxonomy of the pretraining data more than the size. For example, supervised-pretraining on Kinetics gives better performance on both UCF101 and HMDB51 compared to supervised-pretraining on AudioSet (which is $8$ times larger than Kinetics) and ImageNet. One the other hand, XDC performance is less sensitive to the data type, as it implicitly learns the label space rather than depend on a space manually defined by annotators.

**Study 4: Curated vs. uncurated pretraining data.** The overarching goal of self-supervised representation learning is to learn from the massive amounts of unlabeled data. Previous self-supervised methods have pretrained on videos from supervised (curated) datasets (*e.g.*, Kinetics) without using the labels. However, even without using labels, those videos are still biased due to the sampling distribution (*e.g.*, taxonomy of the curated dataset). To this end, we study the effects of self-supervised representation learning from uncurated data. Table 4 compares XDC pretrained on IG-Kinetics (curated, as videos were tag-retrieved) vs. IG-Random (uncurated) using $1M$, $16M$, and $65M$ videos. ***Observations:*** **(I)** Curated pretraining gives better results on UCF101 and HMDB51, while uncurated pretraining is better on ESC50 at large scale. We hypothesize that the bias of IG-Kinetics towards semantics of human actions is the reason behind the positive effect of curation on

Table 4: **Curated vs. uncurated pretraining data.** XDC pretrained on IG-Kinetics (curated) vs. IG-Random (uncurated) using different training set sizes. Uncurated pretraining has better results on ESC at large scale. On UCF and HMDB, the accuracy gap between curated and uncurated pretraining decreases as data size increases.

| Downstream Dataset | Pretraining Size | | | | | | | | |
| | **1M** | | | **16M** | | | **65M** | | |
| | IG-Random | IG-Kinetics | $\Delta$ | IG-Random | IG-Kinetics | $\Delta$ | IG-Random | IG-Kinetics | $\Delta$ |
|---|---|---|---|---|---|---|---|---|---|
| UCF101 | 79.6 | **84.2** | -4.6 | 84.1 | **87.6** | -3.5 | 88.8 | **91.5** | -2.7 |
| HMDB51 | 45.1 | **50.3** | -5.2 | 55.2 | **57.3** | -2.1 | 61.2 | **63.1** | -1.9 |
| ESC50 | 77.8 | **79.5** | -1.7 | **84.3** | 82.5 | +1.8 | **86.3** | 84.8 | +1.5 |

Table 5: **Full finetuning vs. learning `fc`-only.** We compare XDC against the supervised pretrained models (Superv) under two transfer-learning schemes: when models are used as features extractor ('`fc`' column) or as a finetuning initialization ('all' column). XDC fixed features outperform several fully-finetuned Superv models.

| Method | Pretraining Dataset | UCF101 | | HMDB51 | | ESC50 | |
| | | `fc` | all | `fc` | all | `fc` | all |
|---|---|---|---|---|---|---|---|
| Random | None | 6.0±1.0 | 54.5 | 7.5±0.6 | 24.1 | 61.3±2.5 | 54.3 |
| Superv | ImageNet | 74.5 | 79.9 | 42.8 | 44.5 | NA | NA |
| Superv | Kinetics | **89.7** | 90.9 | **61.5** | 58.0 | 79.5 | 82.3 |
| Superv | AudioSet | 80.2 | 84.0 | 51.6 | 53.5 | **88.5** | **90.3** |
| XDC | IG-Random | 80.7 | 88.8 | 49.9 | 61.2 | 84.5 | 86.3 |
| XDC | IG-Kinetics | 85.3 | **91.5** | 56.0 | **63.1** | 84.3 | 84.8 |

UCF101 and HMDB51. However, such bias negatively impacts the performance on ESC50. **(II)** The performance gap between the curated and uncurated pretraining shrinks significantly as we increase the data size. For example, the performance gap on HMDB51 drops from $5.2\%$ to $2.1\%$ and $1.9\%$ when the pretraining size increases from 1M to 16M and 65M videos, respectively. This implies that XDC can learn meaningful representations from truly uncurated data. To the best of our knowledge, XDC is the first self-supervised method to study pretraining on large-scale uncurated video data.

**Study 5: Full finetuning vs. learning `fc`-only.** Here, we study two approaches for transferring XDC to downstream tasks. *Full finetuning*: we finetune all parameters of the pretrained encoder on the downstream task. *Learning `fc`-only*: we fix the pretrained encoder and learn a linear classifier for the downstream task, *i.e.*, a fully-connected (`fc`) layer on top of the frozen features. Table 5 compares XDC with the supervised pretrained approaches under these two transfer-learning schemes. ***Observations:*** **(I)** The accuracy of most pretrained models (fully-supervised or self-supervised) degrades, when used as a fixed feature extractor compared to when they are fully-finetuned on the downstream tasks. Nonetheless, the relative performance of XDC compared to supervised pretrained models stays generally the same when fully vs. `fc`-only finetuned on the downstream task. This suggests that XDC pretraining is useful both as a fixed feature extractor and as a pretraining initialization. **(II)** XDC as a fixed feature extractor outperforms many fully-finetuned supervised models. For example, `fc`-only XDC outperforms, by significant margins, the fully-finetuned supervised AudioSet- and ImageNet-pretrained models on both UCF101 and HMDB51. **(III)** We observe that fully-supervised pretraining, followed by `fc`-only finetuning performs well when the pretraining taxonomy is well aligned with that of the downstream task. For example, pretraining on Kinetics by learning `fc`-only on HMDB51 and UCF101 gives the best performance. This is expected as the label spaces of HMBD51 and UCF101 overlap largely with that of Kinetics. This suggests that fully-supervised pretraining is more taxonomy/downstream-task dependent, while our self-supervised XDC is taxonomy-independent.

# 5 Understanding XDC

What does XDC actually learn? What semantic signals does the algorithm use to train its encoders? Here, we try to answer these questions by inspecting the $k$-means clustering results produced by the last iteration of XDC. Figure 2 visualizes some audio and video clusters learned by XDC when trained on Kinetics. These clusters are the top 2 audio clusters (left) and the top 2 video clusters (right) ranked by purity *w.r.t.* Kinetics action labels. More clusters are presented in Table 6. We observe that the top-purity clusters learned from both modalities exhibit strong semantic coherence. For example, the audio 1st and 8th ranked clusters include concepts related to playing musical instruments that have similar sounds, while the 1st ranked video cluster also groups playing-instrument concepts, but mainly because of their appearance, as the cluster is all about guitars. Other interesting clusters include: grouping by motor-engine sounds (audio #10), by different swimming strokes (video #4), by

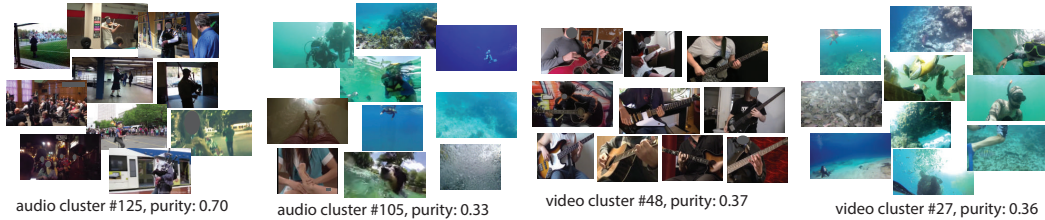

audio cluster #125, purity: 0.70    audio cluster #105, purity: 0.33    video cluster #48, purity: 0.37    video cluster #27, purity: 0.36

Figure 2: **Visualization of XDC clusters on Kinetics videos**. The top-2 audio clusters (left) and video clusters (right) in terms of purity *w.r.t.* the Kinetics labels. Clusters are represented by the 10 closest videos (shown as frames) to their centroid. Interestingly, XDC learned to group "scuba diving" with "snorkeling" (second left, cluster #105) based on audio features and "scuba diving" with "feeding fish" (rightmost, cluster #27) based on visual features. Please refer to our project website for an interactive visualization of all XDC clusters.

Table 6: **XDC clusters.** Top and bottom audio (left) and video (right) XDC clusters ranked by clustering purity *w.r.t.* Kinetics labels. For each cluster, we list the three concepts with the highest purity (given in parentheses).

| # | Kinetics concepts | # | Kinetics concepts |
|---|---|---|---|
| 1 | play bagpipes(0.70), play harmonica(0.04), play violin(0.03) | 1 | play bass guitar(0.37), play guitar(0.16), tap guitar(0.15) |
| 2 | scuba diving(0.33), snorkeling(0.27), feeding fish(0.11) | 4 | swim backstroke(0.21), breast stroke(0.16), butterfly stroke(0.1) |
| 8 | play cello(0.15), play trombone(0.11), play accordion(0.09) | 5 | golf putting(0.18), golf chipping(0.11), golf driving(0.05) |
| 10 | mowing lawn(0.14), driving tractor(0.09), motorcycling(0.06) | 10 | cook chicken(0.11), barbeque(0.07), fry vegetables(0.06) |
| 127 | abseiling(0.01), grooming horse(0.01), milking cow(0.01) | 63 | pull ups(0.01), gymnastics tumbling(0.01), punching bag(0.01) |
| 128 | washing feet(0.01), motorcycling(0.01), headbanging(0.01) | 64 | capoeira(0.01), riding elephant(0.01), feeding goats(0.01) |

different golf shots (video #5), and different cooking activities (video #10). In the bottom-ranked clusters, although the purity *w.r.t.* Kinetics concepts is low, we still find some coherence, mostly at the scene level: a farm setting in audio #127 ("grooming horse", "milking cow") and gym activities in video #63 ("pull ups", "punching bag"). Many other bottom-ranked clusters appear to lack semantic coherence when viewed through the lens of Kinetics labels. However, one of the motivations behind the design of self-supervised methods is precisely to bypass the hand-design of label spaces, which may not be the optimal ones for general representation learning. Our experiments suggest that the label space learned by XDC yields strong and general audio and video features even though it does not align perfectly with the taxonomies of existing datasets.

## 6  State-of-the-art self-supervised learning comparison

**Experimental setup.** Here, training is similar to our ablations except that we re-train our video encoder on the last clustering assignment using 32-frame clips. Then following [28, 63], we finetune on UCF101 and HMDB51 using 32-frame clips for both XDC and the fully-supervised baselines. Inference is similar to our ablations except for using 32-frame clips. For the audio event classification dataset DCASE [59], we follow [28] and extract conv_5 features for 60 uniformly-sampled clips per audio sample and learn a linear SVM. We report the average top-1 accuracy over *all splits*.

**Video action recognition.** Table 7(a) compares XDC pretrained on four large-scale datasets against state-of-the-art self-supervised methods, after finetuning on the UCF101 and HMDB51 benchmarks[2]. We also compare against two fully-supervised methods pretrained on ImageNet and Kinetics. ***Results:*** **(I)** XDC pretrained on IG-Kinetics sets new state-of-the-art performance for self-supervised methods on both benchmarks, outperforming Elo [49] by 1.7% on UCF101 and 1.5% on HMDB51. Moreover, XDC significantly outperforms fully-supervised pretraining on Kinetics: by 1.3% on UCF101 and by 3.8% on HMDB51. **(II)** When directly compared on the same R(2+1)D-18 architecture, XDC pretrained on Kinetics slightly outperforms AVTS [28] by 0.6% on UCF101 and 0.3% on HMDB51. However, when both methods are pretrained on AudioSet, XDC outperforms AVTS with larger margins: by 3.9% on UCF101 and 5.6% on HMDB51. This shows that XDC scales better than AVTS. To further verify that XDC scales better, we pretrained AVTS on AudioSet-240K using R(2+1)D-18 and got 76.9% and 40.7% for UCF101 and HMDB51 on split-1, showing a smaller margin between XDC and AVTS than when both are pretrained on the full AudioSet (cf. Table 3).

**Audio event classification.** Table 7(b) compares XDC pretrained on AudioSet and IG-Random against the state-of-the-art self-supervised methods for audio classification. XDC achieves state-of-the-art performance on DCASE and competitive results on ESC50 with only a 1.1% gap with [54].

Table 7: **State-of-the-art comparison.** We report the average top-1 accuracy over the official splits for all benchmarks. **(a) Video action recognition:** Comparison between XDC with self-supervised and fully-supervised methods on UCF101 and HMDB51. Not only does XDC set new state-of-the-art performance for self-supervised methods, it also outperforms fully-supervised Kinetics and ImageNet pretraining. * For fair comparison with XDC, we report AVTS performance without dense prediction, *i.e.*, we average the predictions of 10 uniformly-sampled clips at inference. † For direct comparison with XDC, we evaluate AVTS using R(2+1)D-18 and 10 uniformly-sampled clips at inference. **(b) Audio event classification:** We compare XDC with self-supervised methods on ESC50 and DCASE. XDC achieves state-of-the-art performance on DCASE.

(a) Video action recognition.

| | Pretraining | | Evaluation | |
|---|---|---|---|---|
| Method | Architecture | Dataset | UCF101 | HMDB51 |
| ClipOrder [75] | R(2+1)D-18 | UCF101 | 72.4 | 30.9 |
| MotionPred [68] | C3D | Kinetics | 61.2 | 33.4 |
| ST-Puzzle [27] | 3D-ResNet18 | Kinetics | 65.8 | 33.7 |
| DPC [17] | 3D-ResNet34 | Kinetics | 75.7 | 35.7 |
| CBT [61] | S3D | Kinetics | 79.5 | 44.6 |
| SpeedNet [4] | S3D | Kinetics | 81.1 | 48.8 |
| AVTS [28]* | MC3-18 | Kinetics | 84.1 | 52.5 |
| AVTS [28]† | R(2+1)D-18 | Kinetics | 86.2 | 52.3 |
| **XDC** (ours) | R(2+1)D-18 | Kinetics | 86.8 | 52.6 |
| AVTS [28]* | MC3-18 | AudioSet | 87.7 | 57.3 |
| AVTS [28]† | R(2+1)D-18 | AudioSet | 89.1 | 58.1 |
| **XDC** (ours) | R(2+1)D-18 | AudioSet | 93.0 | 63.7 |
| MIL-NCE [37] | S3D | HowTo100M | 91.3 | 61.0 |
| ELo [49] | R(2+1)D-50 | YouTube-8M | 93.8 | 67.4 |
| **XDC** (ours) | R(2+1)D-18 | IG-Random | 94.6 | 66.5 |
| **XDC** (ours) | R(2+1)D-18 | IG-Kinetics | **95.5** | **68.9** |
| Fully supervised | R(2+1)D-18 | ImageNet | 84.0 | 48.1 |
| Fully supervised | R(2+1)D-18 | Kinetics | 94.2 | 65.1 |

(b) Audio event classification.

| Method | ESC50 |
|---|---|
| Random Forest [48] | 44.3 |
| Piczak ConvNet [47] | 64.5 |
| SoundNet [2] | 74.2 |
| $L^3$-Net [1] | 79.3 |
| AVTS [28] | 82.3 |
| ConvRBM [54] | **86.5** |
| **XDC** (AudioSet) | 84.8 |
| **XDC** (IG-Random) | 85.4 |

| Method | DCASE |
|---|---|
| RG [50] | 69 |
| LTT [34] | 72 |
| RNH [52] | 77 |
| Ensemble [58] | 78 |
| SoundNet [2] | 88 |
| $L^3$-Net [1] | 93 |
| AVTS [28] | 94 |
| **XDC** (AudioSet) | **95** |
| **XDC** (IG-Random) | **95** |

# 7 XDC for temporal action localization

In this section, we further demonstrate that XDC can be useful beyond video and audio classification. In particular, we employ the recent G-TAD [76] action localization algorithm, where we replace the clip features (originally extracted from a TSN [69] model pretrained on Kinetics) with our XDC features from the R(2+1)D-18 model pretrained on IG-Kinetics or IG-Random. We compare against the features from the R(2+1)D-18 model fully-supervised pretrained on Kinetics. We emphasize that we do not finetune any of the feature extractors used in this experiment. We follow the default hyperparameters setting of G-TAD. Table 8 shows temporal action localization results of G-TAD with different features on THUMOS14 [24] dataset. It reports the mean Average Precision (mAP) results at different temporal Intersection over Union (tIoU) thresholds. Both XDC variants outperform the fully-supervised features across all tIoU thresholds. This confirms the same trend observed in tasks presented in Section 6 and suggests that XDC can be used for other tasks.

Table 8: **Temporal action localization on THUMOS14.** We compare G-TAD algorithm using our XDC features vs. using the fully-supervised Kinetics-pretrained (Superv) features. We report the mean Average Precision (mAP) results at different temporal Intersection over Union (tIoU) thresholds. Both XDC variants outperform the fully-supervised features across all tIoU thresholds.

| | mAP @ tIoU | | | | |
|---|---|---|---|---|---|
| Features Type | 0.3 | 0.4 | 0.5 | 0.6 | 0.7 |
| Superv (Kinetics) | 50.9 | 44.4 | 36.6 | 28.4 | 19.8 |
| XDC (IG-Random) | **51.5** | 44.8 | 36.9 | 28.6 | **20.0** |
| XDC (IG-Kinetics) | **51.5** | **44.9** | **37.2** | **28.7** | **20.0** |

# 8 Conclusion

We presented Cross-Modal Deep Clustering (XDC), a novel self-supervised model for video and audio. XDC outperforms not only existing self-supervised methods but also fully-supervised ImageNet- and Kinetics-pretraining for action recognition. To the best of our knowledge, XDC is the first to show self-supervision outperforming large-scale full-supervision pretraining for action recognition when pretrained on the same architecture and a larger number of uncurated videos.

## Broader Impact Statement

Video has become a commonplace in society. Its uses range from entertainment, to communication and teaching. Thus, the learning of semantic representations of video has broad and far-reaching potential applications. The authors do not foresee major ethical issues associated to this work. However, as the proposed approach is self-supervised, it will learn the inherent properties and structure of the training data. Thus, the learned model may exhibit biases intrinsically present in the data.

## Acknowledgments

The authors thank Mengmeng Xu for his valuable help with the THUMOS14 experiments. The authors appreciate the anonymous NeurIPS reviewers for their constructive feedback. Humam Alwassel was partially supported by the King Abdullah University of Science and Technology (KAUST) Office of Sponsored Research (OSR) under Award No. OSR-CRG2017-3405.

## Footnotes

[2]All XDC pretrained models are publicly released on our project website.

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
