[Supplementary Material]

# Self-Supervised Learning by Cross-Modal Audio-Video Clustering
## — Supplementary Material —

**Humam Alwassel**[1]*
humam.alwassel@kaust.edu.sa

**Dhruv Mahajan**[2]
dhruvm@fb.com

**Bruno Korbar**[2]
bkorbar@fb.com

**Lorenzo Torresani**[2]
torresani@fb.com

**Bernard Ghanem**[1]
bernard.ghanem@kaust.edu.sa

**Du Tran**[2]
trandu@fb.com

[1]King Abdullah University of Science and Technology (KAUST)    [2]Facebook AI
http://humamalwassel.com/publication/xdc

## A    Optimization challenges

In this section, we give the details of the full optimization cycle and discuss differences between the single-modality baseline and our multi-modal models.

**Trivial solutions.** As discussed in [1], SDC may converge to trivial solutions, corresponding to empty clusters or encoder parameterizations, where the classifier predicts the same label regardless of the input. DeepCluster proposes workarounds to tackle these issues, involving reassigning empty cluster centers and sampling training images uniformly over the cluster assignments. While these strategies mitigate the issues, they do not fix the main cause of the problem: SDC learns a discriminative classifier on the *same* input from which it learns the labels. On the other hand, our multi-modal deep clustering models are less prone to trivial solutions because they learn the discriminative classifier on one modality and obtain the labels from a *different* modality. In our training, we never encountered the issue of empty clusters or few-class predictions for any of our multi-modal clustering approaches.

**Initialization and convergence.** Our initial pseudo-labels come from clustering features of randomly-initialized encoders. Such pseudo-labels are "good enough" to capture some weak similarities between the input samples as features from randomly-weighted networks have shown decent results on image and audio classification [6, 7]. Another potential option involves generating the initial pseudo-labels by clustering hand-crafted features, *e.g.* iDT [8] and audio spectrograms. Hand-crafted features capture low-level semantics that may help the encoders learn better or faster. Indeed, in small-scale experiments, we observed that clustering handcrafted features in the initial iteration reduces the number of clustering iterations needed to learn a well-performing encoder. However, we decided to not pursue this further, since these features are computationally expensive to extract and thus are not suitable for large-scale training on millions of examples. Furthermore, handcrafted features may bias the learning to reflect the design choices behind these manually-engineered descriptors.

**Clustering and optimization schedule.** Following previous work [1], we cluster the deep features using the $k$-means algorithm primarily for its desirable properties of efficiency and scalability. The number of $k$-means clusters is a key hyperparameter in our framework. Intuitively, using more clusters makes the pretext task harder, as it increases the number of pseudo-classes the classifier must recognize. On the other hand, the diversity of samples to cluster effectively dictates the maximum $k$, for which the grouping is still sensible. Taking into account these factors, we explore the effects of $k$ in our ablation study in Subsection 4.2 of the main manuscript. Another important hyperparameter of

Table 1: **Training parameter definitions.** The abbreviations and descriptions of each training parameters.

| Abv. | Name | Description |
|---|---|---|
| es | epoch size | The total number of examples the model trains on in one epoch. |
| bs | batch size | The size of a mini-batch. |
| lr | base lr | The initial learning rate. |
| we | warmup epoch | The number of epochs used for warmup [4]. |
| se | step epoch | Every se epochs, the learning rate |
| $\gamma$ | lr decay | is decayed by multiplying with $\gamma$. |
| te | total epoch | The training lasts for te epochs. |
| wd | weight decay | The weight decay used in SGD. |
| e-stop | early stop | Stop training when validation loss is increased in 3 consecutive epochs. |

Table 2: **Pretraining parameters.** We use early-stopping for Kinetics and AudioSet since we observe some overfiting on the pretext tasks. For the last iteration of XDC on IG-Kinetics and IG-Random, we pretrain XDC 3x longer (iteration denoted as IG-Kinetics* and IG-Random* in this table). $\gamma$ is set to 0.01 for all settings.

| method | dataset | es | bs | lr | we/se/te | wd | e-stop |
|---|---|---|---|---|---|---|---|
| Superv | Kinetics | 1M | 32 | 0.01 | 10/10/45 | $10^{-4}$ | no |
| Superv | AudioSet | 2M | 32 | 0.04 | 10/20/45 | $10^{-5}$ | no |
| All DCs | Kinetics | 1M | 32 | 0.01 | 10/10/30 | $10^{-4}$ | yes |
| All DCs | AudioSet | 2M | 32 | 0.01 | 10/10/45 | $10^{-4}$ | yes |
| All DCs | IG-Kinetics & IG-Random | 10M | 32 | 0.01 | 1/3/10 | $10^{-4}$ | no |
| All DCs | IG-Kinetics* & IG-Random* | 10M | 32 | 0.01 | 0/9/30 | $10^{-4}$ | no |

our framework is the number of training epochs for the encoders, before re-clustering the learned features. DeepCluster re-clusters after each epoch, which is an expensive design choice when scaling to millions of training samples. Thus, we choose to fix the pseudo-labels and train the encoders until the validation loss for predicting the pseudo-labels saturates. Then, we re-cluster the newly learned features, reassign pseudo-labels, reset the classification layer, and repeat the same process. We find this strategy to be more efficient, as it reduces the number of times we need to invoke $k$-means.

## B    Learning using audio rather than text from ASR

We note that while our approach was demonstrated by leveraging audio, the method is general and is easy to adapt to other modalities, including text. While video and text are semantically correlated, audio and video are temporally correlated. Thus, these two form of correlations are likely to provide different forms of self-supervision, potentially leading to further gains when used in combination. A disadvantage of text from ASR is that it is only available for videos with speech. Audio provides information about environmental sounds beyond speech (*e.g.* walking steps, playing guitar, and dog barking) and allows us to train on uncurated datasets of arbitrary Web videos, as we demonstrated with IG-Random.

## C    Hyperparameters and training details

**Training**. We train our models using caffe2 with distributed SGD on a GPU cluster, and employ the warmup scheme proposed in [4]. The main training parameters are presented in Table 1. We note that the epoch size can be different from the actual number of videos. This is because the total number of clips the model sees during training (with temporal jittering) can be larger than the number of videos.

**Pretraining parameters**. We pretrain XDC and other baselines using the parameters described in Table 2. Early stopping is used for pretraining on small datasets such as Kinetics [5] and AudioSet [2] to stop before the model starts overfitting on the pretext task. For IG-Kinetics [3] and IG-Random, we do not observe overfitting. We pretrain XDC on IG-Kinetics and IG-Random longer in the last deep clustering iteration (denoted as IG-Kinetics* and IG-Random* in Table 2). When pretraining our R(2+1)D on longer clips (*e.g.* 32 frames), due to the GPU memory limit, we reduce the mini-batch size to 8 (instead of 32) and the base learning rate to 0.0025 (instead of 0.01).

Table 3: **Finetuning parameters.** Different pretraining methods have different ranges of optimal base learning rate when finetuning on downstream tasks. Thus, we cross-validate all methods with the same set of base learning rates and report the best result for each method. $\gamma$ is set to $0.01$ for all settings.

| dataset | es | bs | we/se/te | wd | e-stop |
|---------|------|----|----------|-------|--------|
| HMDB51  | 40K  | 32 | 2/2/8    | 0.005 | no     |
| UCF101  | 106K | 32 | 2/2/8    | 0.005 | no     |
| ESC50   | 20K  | 32 | 2/2/8    | 0.005 | no     |

Table 4: **Finetuning base learning rates.** For a fair comparison, we cross-validate all pretraining methods with the same set of base learning rates. We report the best finetuning result for each method. Learning FC-only benefits from cross-validation with a wider range of base learning rates.

| Setup   | Base learning rates |
|---------|---------------------|
| Full    | $0.001, 0.002, 0.004, 0.006, 0.008, 0.01$ |
| FC only | $0.001, 0.002, 0.004, 0.006, 0.008, 0.01, 0.02, 0.04$ |

Table 5: **XDC using a different backbone.** We present the results of XDC on a different backbone, ResNet3D-18, for the visual encoder. We compare XDC pretrained on Kinetics vs. the two baselines: Scratch and fully-supervised Kinetics-pretraining (Superv) for the same backbone. We report the top-1 accuracy on split-1 of each dataset.

| Method | UCF101 | HMDB51 | ESC50 |
|--------|--------|--------|-------|
| Scratch (ResNet3D-18) | 60.1 | 25.7 | 54.3 |
| Superv (ResNet3D-18)  | 87.5 | 54.5 | 82.3 |
| XDC (ResNet3D-18)     | 68.0 | 36.3 | 75.5 |

**Finetuning parameters**. We provide finetuning hyperparameters in Table 3. Different pretraining methods may have different optimal base learning rate when finetuned on downstream tasks. Thus to make a fair comparison, we cross-validate the finetuning using the same set of base learning rates (presented in Table 4) and report the best result for each pretraining method. As we observed that higher learning rates tend to be beneficial when learning FC-only, we use a wider set of learning rates to cross-validate FC-only models. As done during pretraining, when finetuning R(2+1)D on longer clips (*i.e.* 32 frames), we reduce the mini-batch size to $8$ and reduce the base learning rate to $1/4$ of its original rate.

## D   XDC using a different backbone architecture

We pretrain XDC on Kinetics with ResNet3D-18 as the visual backbone and keep the same audio encoder (ResNet-18). The results are compared with those of baselines in Table 5. XDC with the ResNet3D-18 backbone outperforms the training from scratch baseline by good margins on three downstream tasks.

## E   Additional qualitative results

**XDC clusters.** Tables 6 and 7 present the top and bottom 10 audio and video clusters learned with XDC on Kinetics, ranked by their purity with respect to Kinetics labels. We list the 5 most frequent concepts of each cluster.

**XDC filters.** Figure 1 visualizes and compares `conv_1` spatial and temporal filters of R(2+1)D learned by self-supervised XDC pretraining on IG-Kinetics versus fully-supervised pretraining on Kinetics. We observe some differences in both spatial and temporal filters between XDC and fully-supervised pretraining. In particular, XDC learns a more diverse set of motion filters.

## Footnotes

*Work done during an internship at Facebook AI

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

Table 6: **XDC audio clusters.** Top and bottom 10 XDC audio clusters ranked by clustering purity *w.r.t.* Kinetics labels. For each, we list the 5 concepts with the highest purity (given in parentheses).

| # | Kinetics concepts |
|---|---|
| 1 | playing bagpipes(0.70), playing 2harmonica(0.04), playing violin(0.03), playing accordion(0.02), marching(0.01) |
| 2 | scuba diving(0.33), snorkeling(0.27), feeding fish(0.11), canoeing or kayaking(0.02), jumping into pool(0.02) |
| 3 | playing cymbals(0.21), playing drums(0.17), marching(0.03), air drumming(0.02), drumming fingers(0.02) |
| 4 | passing American football(0.17), play kickball(0.06), catching or throwing softball(0.05), kick field goal(0.02), sled dog racing(0.02) |
| 5 | presenting weather forecast(0.17), playing poker(0.05), testifying(0.03), tying knot (not on a tie)(0.02), golf putting(0.02) |
| 6 | hurling (sport)(0.17), swimming backstroke(0.05), skiing slalom(0.04), vault(0.03), ski jumping(0.02) |
| 7 | presenting weather forecast(0.15), news anchoring(0.05), filling eyebrows(0.02), braiding hair(0.02), tossing salad(0.02) |
| 8 | playing cello(0.15), playing trombone(0.11), playing accordion(0.09), playing harp(0.07), playing clarinet(0.06) |
| 9 | playing recorder(0.14), playing violin(0.12), playing trumpet(0.08), playing harmonica(0.07), tapping guitar(0.06) |
| 10 | mowing lawn(0.14), driving tractor(0.09), motorcycling(0.06), blowing leaves(0.04), water skiing(0.04) |
| 119 | side kick(0.02), front raises(0.01), dunking basketball(0.01), smoking(0.01), high kick(0.01) |
| 120 | clay pottery making(0.02), crawling baby(0.02), brushing teeth(0.01), playing harmonica(0.01), eating spaghetti(0.01) |
| 121 | pushing cart(0.01), hula hooping(0.01), high kick(0.01), blowing out candles(0.01), bench pressing(0.01) |
| 122 | shot put(0.01), feeding birds(0.01), squat(0.01), push up(0.01), high jump(0.01) |
| 123 | opening present(0.01), petting cat(0.01), pushing cart(0.01), washing dishes(0.01), punching bag(0.01) |
| 124 | trimming or shaving beard(0.01), petting cat(0.01), front raises(0.01), massaging back(0.01), tai chi(0.01) |
| 125 | feeding birds(0.01), tobogganing(0.01), riding elephant(0.01), feeding goats(0.01), jumping into pool(0.01) |
| 126 | climbing tree(0.01), writing(0.01), archery(0.01), brushing hair(0.01), shining shoes(0.01) |
| 127 | abseiling(0.01), grooming horse(0.01), milking cow(0.01), feeding goats(0.01), juggling balls(0.01) |
| 128 | washing feet(0.01), motorcycling(0.01), headbanging(0.01), cheerleading(0.01), krumping(0.01) |

Table 7: **XDC video clusters.** Top and bottom 10 XDC video clusters ranked by clustering purity *w.r.t.* Kinetics labels. For each, we list the 5 concepts with the highest purity (given in parentheses).

| # | Kinetics concepts |
|---|---|
| 1 | playing bass guitar(0.37), playing guitar(0.16), tapping guitar(0.15), strumming guitar(0.09), playing ukulele(0.09) |
| 2 | scuba diving(0.36), snorkeling(0.32), feeding fish(0.10), diving cliff(0.02), jumping into pool(0.02) |
| 3 | presenting weather forecast(0.26), playing poker(0.10), news anchoring(0.05), testifying(0.03), giving or receiving award(0.02) |
| 4 | swimming backstroke(0.21), swimming breast stroke(0.16), swimming butterfly stroke(0.10), play ice hockey(0.04), jump into pool(0.04) |
| 5 | golf putting(0.18), golf chipping(0.11), golf driving(0.05), hitting baseball(0.03), archery(0.03) |
| 6 | hurling (sport)(0.17), passing American football (in game)(0.06), skiing slalom(0.04), playing ice hockey(0.03), vault(0.03) |
| 7 | filling eyebrows(0.13), braiding hair(0.05), massaging back(0.05), curling hair(0.05), dying hair(0.03) |
| 8 | playing cello(0.12), playing harp(0.12), playing trombone(0.06), playing piano(0.06), playing accordion(0.05) |
| 9 | windsurfing(0.12), jetskiing(0.10), water skiing(0.09), surfing water(0.08), kitesurfing(0.06) |
| 10 | cooking chicken(0.11), barbequing(0.07), frying vegetables(0.06), cooking sausages(0.04), making pizza(0.04) |
| 55 | yoga(0.02), folding napkins(0.02), doing nails(0.02), cutting watermelon(0.01), writing(0.01) |
| 56 | eating spaghetti(0.02), making pizza(0.02), brushing teeth(0.02), blowing out candles(0.02), reading book(0.02) |
| 57 | answering questions(0.02), tai chi(0.02), dancing ballet(0.02), dunking basketball(0.02), sign language interpreting(0.01) |
| 58 | trimming or shaving beard(0.02), barbequing(0.02), kissing(0.02), dining(0.01), playing poker(0.01) |
| 59 | punching bag(0.02), blowing out candles(0.02), pumping fist(0.02), dancing gangnam style(0.02), opening present(0.01) |
| 60 | feeding goats(0.02), blowing out candles(0.02), milking cow(0.02), arm wrestling(0.02), finger snapping(0.02) |
| 61 | air drumming(0.02), pumping fist(0.02), pushing cart(0.02), brushing teeth(0.02), eating ice cream(0.01) |
| 62 | clean and jerk(0.01), robot dancing(0.01), bench pressing(0.01), side kick(0.01), punching bag(0.01) |
| 63 | pull ups(0.01), gymnastics tumbling(0.01), punching bag(0.01), cracking neck(0.01), eating ice cream(0.01) |
| 64 | capoeira(0.01), riding elephant(0.01), feeding goats(0.01), feeding birds(0.01), crawling baby(0.01) |

[3] Deepti Ghadiyaram, Matt Feiszli, Du Tran, Xueting Yan, Heng Wang, and Dhruv Mahajan. Large-scale weakly-supervised pre-training for video action recognition. In *CVPR*, 2019. 2

[4] Priya Goyal, Piotr Dollár, Ross Girshick, Pieter Noordhuis, Lukasz Wesolowski, Aapo Kyrola, Andrew Tulloch, Yangqing Jia, and Kaiming He. Accurate, large minibatch SGD: training imagenet in 1 hour. *arXiv preprint arXiv:1706.02677*, 2017. 2

[5] Will Kay, Joao Carreira, Karen Simonyan, Brian Zhang, Chloe Hillier, Sudheendra Vijayanarasimhan, Fabio Viola, Tim Green, Trevor Back, Paul Natsev, Mustafa Suleyman, and Andrew Zisserman. The kinetics human action video dataset. *CoRR*, abs/1705.06950, 2017. 2

[6] Jordi Pons and Xavier Serra. Randomly weighted cnns for (music) audio classification. In *ICASSP*, 2019. 1

[7] Andrew M Saxe, Pang Wei Koh, Zhenghao Chen, Maneesh Bhand, Bipin Suresh, and Andrew Y Ng. On random weights and unsupervised feature learning. In *ICML*, 2011. 1

[8] Heng Wang and Cordelia Schmid. Action recognition with improved trajectories. In *ICCV*, 2013. 1

a) conv1 spatial and temproal filtes learned by Kinetics fully supervision.

b) conv1 spatial and temporal filters learned by IG65M self-supervised XDC.

Figure 1: **R(2+1)D filters learned with self-supervised XDC vs. fully-supervised training.** (a) R(2+1)D `conv_1` filters learned by fully-supervised training on Kinetics. (b) The same filters learned by self-supervised XDC pretraining on IG-Kinetics. XDC learns a more diverse set of temporal filters compared to fully-supervised pretraining.