[Reviews · NeurIPS 2020]

Review 1

Summary and Contributions: This paper extends the self-supervised image representation learning algorithm DeepCluster [4] to work with audiovisual inputs (encoded by 2D and 3D ResNets respectively). It compares three direct adaptation of the DeepClustering algorithm: multi-head, concatenation and cross-modal, and found that the cross-modal version performs the best. The authors also demonstrated that given large enough data, a self-supervised pretraining is able to outperform fully-supervised pretraining on a smaller dataset (65M v.s. 240K).

Strengths: + Self-supervised representation learning has drawn lots of attention recently, yet learning from videos is comparably less explored, probably due to the computation cost. Work on learning from huge amount of videos would be interesting sharing with the community. + The paper provides extensive ablation studies on different variants of multimodal DeepClustering, the impact of hyperparameters, pretraining data size and type, which would be valuable for researchers working on such topics. + The observation that cross-modal variant is significantly better than the other two alternatives is interesting and worth digging into.

Weaknesses: - Despite the extensive empirical evaluations, the three multimodal variants as proposed by the paper are direct extensions of the DeepCluster algorithm [4]. The main contributions appear to be (1) a working pipeline which demonstrates that variants of DeepCluster works with video and audio encoders; (2) scaling up the training to extremely large datasets. While both contributions are interesting, they appear to me to be less relevant to the audience of NeurIPS. - I would appreciate if the authors could come up with an explanation / conjecture why crossmodal outperforms single-modal; and when XDC outperforms CDC and MDC. It would also be great if such conjectures are accompanied with empirical evaluations on more diverse tasks than the three classification datasets. That would help the audience understand when to apply the XDC variant of DeepCluster (e.g. is it specific to audio and visual in videos, or is it more general?), and have a nice intuition on why it is so effective, thus can further apply the method on other multimodal tasks. - Minor: the claim that "self-supervised video representation learning outperforms large-scale fully-supervised pretraining for action recognition" is a bit misleading to me as the self-supervised approach requires one or two orders of magnitude larger datasets.

Correctness: The empirical methodology appears to be detailed and correct to me.

Clarity: The paper is very clearly written.

Relation to Prior Work: The related work section offers a nice overview of previous contributions. The authors might be interested in and consider discussing: [a] Self-Supervised Learning of Video-Induced Visual Invariances. (Video representation learning from temporal invariance) [b] Learning Video Representations using Contrastive Bidirectional Transformer. (Video representation learning by contextualized prediction) [c] End-to-End Learning of Visual Representations from Uncurated Instructional Videos (Video representation learning from unlabeled, instructional videos) [d] Contrastive multiview coding (Representation learning from different views of the same data. Audiovisual learning is a special case)

Reproducibility: Yes

Additional Feedback:


Review 2

Summary and Contributions: This paper presents a clustering-based self-supervised learning algorithm from the video and audio sources. The paper extends the existing work on the single modality (i.e., image) into multi-modality while clustering-based assignments from one modality are used as pseudo labels in the other modality. Furthermore, the authors achieve a higher performance even compared to fully supervised models.

Strengths: 1. Powerful performance. The method is the first method that provides better a pre-training model even compared to a fully supervised pre-training one, which will benefit the following research in videos. 2. Transferability of clustering assignments The obtained clustering assignments can be transferred to other datasets. This property is a unique characteristic because other self-supervised methods can transfer only pre-trained weights. Also, it can reduce the cost to annotate data from large-scale videos on the web.

Weaknesses: 1. Missing baseline. It is unclear why XDC is better than the others (i.e., MDC and CDC). To check learning a model using cluster assignments from another model (not from different modality) is important, training two networks from single modality while guiding each other would be an important baseline. 2. Generalizability of the method The generalization capability of the method over diverse backbone networks is not demonstrated. If the experimental results using a different or larger backbone network (e.g., S3D or R(2+1)D-34) can be included, it will strengthen the paper quality. I think the authors employed a backbone network of R(2+1)D-18 due to huge training time. The authors can use cluster assignments from R(2+1)D-18 to train larger backbones and compare performances with fully supervised ones. This will show generalizability as well as effective of cluster assignments.

Correctness: All correct.

Clarity: Clearly well written and easy to understand. However, to correclty reproduce the method, readers should read the previous work.

Relation to Prior Work: The paper is highly related to the previous one, however, the authors clearly describe the difference and validate the effectiveness of the method. Some references missing. Like the paper, recently, self-supervised learning methods on multi-modal sources in videos are proposed, for example, MIL-NCE [A] and CBT [B]; they are different compared to the paper in respect to relying on text from ASR instead of audio signals. Considering the fact that language sometimes directly contain semantics and can be used as labels, using text obatined by ASR would be more powerful guidance to learn visual representation. Please discuss why we focus on self-supervised learning using audio and video rather than text. [A] A. Miech et. al., End-to-End Learning of Visual Representations from Uncurated Instructional Videos, CVPR, 2020 [B] C. Sun et. al., Learning Video Representation Using Constrastive Bidirectional Transformer, arXiv preprint arXiv:1906.05743, 2019

Reproducibility: Yes

Additional Feedback: As an alternative method towards better initialization of pseudo labels, instead of relying on clustering features of randomly initialized encoders, how about pre-training an encoder using other self-supervised methods? How long training a model on IG-Random or IG-Kinetics takes?


Review 3

Summary and Contributions: This paper proposes a self-supervised pretraining method that learns representations performing clustering within domains and predicting cluster labels across domains. It achieves state of the art results in multiple benchmark datasets both in action recognition and sound classification. It even outperforms supervised pre-training methods with the help of very large scale uncurated datasets.

Strengths: + The successfully extends single modality (vision) deep clustering approaches to multiple modalities (vision and sound). + It proposes and evaluates three variants of the proposed method which covers a wide range of potential architectural choices. It is interesting to note that XDC works better than the other 2 variants. + The authors do a very good job on evaluating the learned representation on a diverse set of problems and datasets together with insightful ablation studies. + They achieve state of the art results on UCF101, HMDB51 and ESC50 datasets. + They even outperform supervised pre-training on UCF101 and HMDB51, though they use much larger scale of unlabelled datasets. + Despite the fact that they have plenty of results, it is presented in a well structured format which makes it quite easy to follow. + Cluster visualisations are helpful.

Weaknesses: - Some well explained intuitions or, if possible, theoretically grounded explanations for why XDC works better than MDC and CDC would further improve the quality of this great paper.

Correctness: Appears to be so! The paper has a very sound evaluation methodology which covers many potential ways of evaluating the learned representation.

Clarity: Yes, it is very well written!

Relation to Prior Work: The difference from the existing work is clearly described.

Reproducibility: Yes

Additional Feedback: The references are already quite comprehensive but the paper below is also relevant and could be included: See, hear, and read: Deep aligned representations Y Aytar, C Vondrick, A Torralba arXiv preprint arXiv:1706.00932 Update after rebuttal: I read the reviews and the rebuttal. In general I am quite positive about this paper mainly because they show that with cross-modal audio-visual self-supervision one can reach and surpass the performance of supervised features for action recognition. They also do a great job on thorough experimentation and structured presentation of the paper and the results. Though I am also curios about why XDC performs better than the other two variants, for now I'll accept the empirical justifications which are clearly presented with extensive experiments. The additional results on action localisation is also quite encouraging. I also like R2's suggestion on alternative experiments to figure out the main reason why XDC performing better than the others. Perhaps similar analysis can be performed for CDC and MDC as well. I'll keep my recommendation as is.


Review 4

Summary and Contributions: - Paper proposes a cross modal clustering task for self-supervision. Clustering labels from one domain are used to supervise the other domain with the reasoning that correlations and complementary nature of the two streams can help for self-supervision. - Their approach to learn on unlabeled data outperforms models pre-trained with Kinetics on HMDB and UCF101 - claims to be the first method to do so - propose 3 approaches to solve the task. One where the encoder predicts the cluster assignments of both modalities, one where clustering of both modalities is done together by concatenating and finally one where the cluster signal of one is exclusively used for only the other modality. - Interestingly they find that the cross-modal self supervision performs the best.

Strengths: - I like the idea and motivation behind using one domain to generate 'supervisory' signals for the other domain. It is a good extension to DeepCluster in the multi modal setting. - The observation that cross-modal alone works better than within-modality is very interesting - Authors show results on pre-training with large-scale uncurated datasets thus demonstrating the actual usefulness of the approach. Making use of these uncurated datasets to pre-train gives them performance that is better than pre-training on Kinetics. - The authors perform extensive ablation study to justify design choices. - I like the discussion on label-space for the downstream task - It is a simple idea that works - can be possibly adapted widely and and other interesting ideas can be built upon it.

Weaknesses: Some more analysis on why the proposed approach works would be beneficial. The paper does talk about what XDC learns, but why does the proposed approach work better than in-domain/single domain clustering and why does XDC work better than the other variants proposed? An experiment with AVTS + IG-Random and IG-Kinetics would help gauge how the proposed method compares to previous methods on very large datasets. This will help strengthen the claim on L287-288

Correctness: The claims in the paper are sound and the empirical methodology is correct.

Clarity: The paper is well written and easy to follow. The figures and tables are well explained.

Relation to Prior Work: The related work section is well written. The authors discuss how this work differs from previous papers and the papers they build upon.

Reproducibility: Yes

Additional Feedback: Please check some questions in the weakness section. Post rebuttal: I have read the reviews and the rebuttal submitted by the authors. I too agree with the other reviewers that the method is an extension of DeepCluster[4], but I think it would be an interesting paper for the community to read and build upon. The presented experiments, ablations, and the simplicity of the approach could be useful. The paper's discussion on label-space for video based self-supervised learning tasks is also interesting. That said, I believe the reasoning behind why the proposed approach performs better than CDC/MDC requires more exploration and analysis. I am not too convinced with the reasoning provided. I agree with R1’s point that the paper lacks enough discussion on which settings/tasks this method could be applied on. Though their results for temporal action localization as mentioned in the rebuttal seem promising.

[Author Response · NeurIPS 2020]

**Why XDC outperforms CDC and MDC?** [all reviewers]. We have shown in Study I (Table 1) that XDC quantitatively outperforms both CDC and MDC on three downstream tasks. We provide the following intuition on why XDC is the best of the three models. XDC groups samples together when they are similar in one of the two modalities (video to supervise the audio encoder, audio to supervise the visual encoder). Instead, CDC groups samples together only if they are similar according to both the audio *and* the video modality (to supervise both encoders). Thus, XDC visual and audio clusters allow for more diversity than those of CDC. We hypothesize that this diversity allows XDC to learn richer representations, which translates into better performance on the downstream tasks. Also, recent work [A1] has shown that models trained on different modalities learn and generalize at different speeds, and that training them jointly (as done in MDC which uses two-modality heads) is sub-optimal. We believe that this could contribute to MDC performing worse than XDC, which optimizes for each modality independently.

**Cross-modality vs. single-modality** [R1, R2, R4]. We thank R2 for suggesting the insightful baseline corresponding to training XDC with the two encoders defined on the same modality (either visual or audio). Table A compares this baseline to SDC. It can be seen that the same-modality-XDC baselines perform similarly to SDC and are 8-12% worse than multi-modal-XDC. This suggests that cross-modality provides a superior supervisory signal for self-supervised learning and that multi-modal-XDC is the best model not because of its optimization strategy but rather because of the use of the other modality for pseudo-labeling.

**XDC using a different backbone** [R2]. We pretrain XDC on Kinetics with ResNet3D-18 as the visual backbone and keep the same audio encoder. The results are compared with those of baselines in Table B. XDC with the ResNet3D-18 backbone outperforms the training from scratch baseline by good margins on three downstream tasks.

**XDC for other tasks** [R1]. Table C provides the results of transferring XDC to the task of temporal action localization on THUMOS14 dataset. We employ the recent G-TAD [A2] algorithm, where we replace the clip features (originally extracted from a TSN model pretrained on Kinetics) with XDC features from the R(2+1)D-18 model pretrained on IG-Kinetics or IG-Random. We compare against the features from the R(2+1)D-18 model fully-supervised pretrained on Kinetics. We do not finetune any of the feature extractors used in this experiment. Both XDC variants outperform the fully-supervised features across all temporal Intersection over Union (tIoU) thresholds. This confirms the same trend observed in the tasks discussed in the paper and suggests that XDC can also be used for other tasks.

**Learning using audio rather than text from ASR** [R2]. We note that while our approach was demonstrated by leveraging audio, the method is general and is easy to adapt to other modalities, including text. While video and text are semantically correlated, audio and video are temporally correlated. Thus, these two form of correlations are likely to provide different forms of self-supervision, potentially leading to further gains when used in combination. A disadvantage of text from ASR is that it is only available for videos with speech. Audio provides information about environmental sounds beyond speech (*e.g.* walking steps, playing guitar, and dog barking) and allows us to train on uncurated datasets of arbitrary Web videos, as we demonstrated with IG-Random.

**AVTS pretrained on IG-Kinetics and IG-Random** [R4]. Training on such large datasets is expensive and unfortunately cannot be done within the short rebuttal period. However, we extensively compared XDC against AVTS (Section 6) pretrained on Kinetics, AudioSet-240K, and AudioSet using the same backbone. These results suggest that XDC outperforms AVTS consistently under the same settings on UCF101 and HMDB51.

**Other comments**. We thank R2 for suggesting an alternative pseudo-labeling initialization method. We will investigate this approach. Training on IG-Kinetics or IG-Random takes about 360 hours on 160 V100 GPUs. We will add the suggested references (by R1, R2, R3) to the final version and adjust the claim on using more data (by R1). We truly appreciate the constructive feedback from all reviewers.

# References

[A1] Wang *et al.* What makes training multi-modal classification networks hard? In *CVPR*, 2020.

[A2] Xu *et al.* G-TAD: Sub-graph localization for temporal action detection. In *CVPR*, 2020.

Table A: XDC using two encoders of the same modality. We use Kinetics for pretraining and report the top-1 accuracy on split-1 of each dataset.

| Method | UCF101 | HMDB51 | ESC50 |
|---|---|---|---|
| XDC-visual-encoders | 61.3 | 30.5 | N/A |
| XDC-audio-encoders | N/A | N/A | 66.0 |
| SDC | 61.8 | 31.4 | 66.5 |
| XDC | 74.2 | 39.0 | 78.0 |

Table B: XDC with ResNet3D-18 pretrained on Kinetics. We compare against the baselines: Scratch and fully-supervised pretraining (Superv) on the same backbone.

| Method | UCF101 | HMDB51 | ESC50 |
|---|---|---|---|
| Scratch | 60.1 | 25.7 | 54.3 |
| Superv | 87.5 | 54.5 | 82.3 |
| XDC | 68.0 | 36.3 | 75.5 |

Table C: Temporal action localization on THUMOS14. We compare G-TAD [A2] algorithm using XDC features vs. using fully-supervised pretrained (Superv) features.

| Method | mAP @ tIoU | | | | |
|---|---|---|---|---|---|
| | 0.3 | 0.4 | 0.5 | 0.6 | 0.7 |
| Superv (Kinetics) | 50.9 | 44.4 | 36.6 | 28.4 | 19.8 |
| XDC (IG-Random) | **51.5** | 44.8 | 36.9 | 28.6 | **20.0** |
| XDC (IG-Kinetics) | **51.5** | **44.9** | **37.2** | **28.7** | **20.0** |

[Meta-Review · NeurIPS 2020]

The reviewers generally agree this paper has great execution, a great idea, and great results. The reviewers noted the impact that self-supervised learning on video can have, which has been less explored than the image counterpart. The reviewers also praised the strong empirical results, which will be of high interest to the community. The clear visualizations and strong ablation experiments further support the claims in the paper. Congratulations!